Leveraging sentiment analysis of food delivery services reviews using deep learning and word embedding

http://orcid.org/0000-0003-1456-7377 Mustafa Dheya 1 dheya@hu.edu.jo
Khabour Safaa M. 2
http://orcid.org/0000-0003-3180-3861 Al-kfairy Mousa 3
Shatnawi Ahmed 4
1 Department of Computer Engineering, Faculty of Engineering, The Hashemite University , Zarqa , Jordan
2 Department of Information Systems, Yarmouk University , Irbid , Jordan
3 College of Technological Innovation, Zayed University , Abu Dhabi , United Arab Emirates
4 Department of Software Engineering, Jordan University of Science and Technology , Irbid , Jordan
Aydın Doğan
Electronic publication date: 2025 Feb 19
Publication date: 2025
Volume: 11
Electronic Location ID: e2669
Received 2024 Sep 16; Accepted 2025 Jan 6
Copyright: © 2025 Mustafa et al.
Copyright year: 2025
Copyright holder: Mustafa et al.
License: This is an open access article distributed under the terms of the Creative Commons Attribution License, which permits unrestricted use, distribution, reproduction and adaptation in any medium and for any purpose provided that it is properly attributed. For attribution, the original author(s), title, publication source (PeerJ Computer Science) and either DOI or URL of the article must be cited.
License URL: https://creativecommons.org/licenses/by/4.0/

Keywords: NLP, Word embedding, Consumer sentiment, Food delivery services, Deep learning, Arabic sentiments resources, Arabic dialects, CNN, LSTM-CNN, BiLSTM

Funding: Zayed University RIF R22085 This research was supported by the Zayed University RIF grant activity code R22085. There was no additional external funding received for this study. The funders had no role in study design, data collection and analysis, decision to publish, or preparation of the manuscript.

==============================
Companies that deliver food (food delivery services, or FDS) try to use customer feedback to identify aspects where the customer experience could be improved. Consumer feedback on purchasing and receiving goods via online platforms is a crucial tool for learning about a company’s performance. Many English-language studies have been conducted on sentiment analysis (SA). Arabic is becoming one of the most extensively written languages on the World Wide Web, but because of its morphological and grammatical difficulty as well as the lack of openly accessible resources for Arabic SA, like as dictionaries and datasets, there has not been much research done on the language. Using a manually annotated FDS dataset, the current study conducts extensive sentiment analysis using reviews related to FDS that include Modern Standard Arabic and dialectal Arabic. It does this by utilizing word embedding models, deep learning techniques, and natural language processing to extract subjective opinions, determine polarity, and recognize customers’ feelings in the FDS domain. Convolutional neural network (CNN), bidirectional long short-term memory recurrent neural network (BiLSTM), and an LSTM-CNN hybrid model were among the deep learning approaches to classification that we evaluated. In addition, the article investigated different effective approaches for word embedding and stemming techniques. Using a dataset of Modern Standard Arabic and dialectal Arabic corpus gathered from Talabat.com, we trained and evaluated our suggested models. Our best accuracy was approximately 84% for multiclass classification and 92.5% for binary classification on the FDS. To verify that the proposed approach is suitable for analyzing human perceptions in diversified domains, we designed and carried out excessive experiments on other existing Arabic datasets. The highest obtained multi-classification accuracy is 88.9% on the Hotels Arabic-Reviews Dataset (HARD) dataset, and the highest obtained binary classification accuracy is 97.2% on the same dataset.

Introduction

In response to the COVID-19 lockdowns and restrictions on people’s movements, food delivery services (FDSs) have reacted to the increasing demand for online food delivery markets by providing a variety of restaurant options for a pleasant experience at home and work. Some examples of third-party marketplace apps that employ an aggregator business model and handle all delivery functions are UberEATS, Menulog, and Talabat (the Arabic word for orders). In reaction to an increase in orders and evaluations, most organizations aim to increase client satisfaction by utilizing data to locate sections that want more improvements (Arora, Srivastava & Bansal, 2020; Wu & Chang, 2020; Adak, Pradhan & Shukla, 2022).

Sentiment analysis (SA) is a classification method that uses textual material to determine people’s opinions and moods (Li et al., 2023; Hassonah et al., 2020). With the ability to learn several layers of abstractions or features of the data and generate cutting-edge prediction results, deep learning has become a potent machine learning tool. In addition to the prominence of deep learning in numerous other fields of application, deep learning has become more prevalent in SA in the last few years (Zhang, Wang & Liu, 2018).

The distinctive qualities of the Arabic language have been investigated in several surveys that have looked at a wide range of research, cutting-edge sentiment analysis techniques, and readily accessible Arabic sentiment resources (Guellil, Azouaou & Mendoza, 2019; Abo, Raj & Qazi, 2019; Oueslati et al., 2020). According to Guellil, Azouaou & Mendoza (2019), the accuracy of the SA approach for Arabic and its dialects has been severely hampered by a lack of resources (Al-Moslmi et al., 2018; Oueslati et al., 2020). They thus concentrated on creating a feeling corpus and lexicon. The language variation among dialects is the reason why most SA methods did not function in the Arabic internet surroundings, as demonstrated by Oueslati et al. (2020). It was suggested that word-based SA be replaced by concept-based SA.

In addition to the usual issues with SA, the literature claims that Arabic variations and morphology present extra challenges. Right-to-left writing is used in the Semitic language of Arabic. Depending on where they appear in the word, Arabic characters can be transcribed in several ways. For example, the letter (kaf) might occur at the beginning of a word (, kahf, cave), in the middle of a word (, maktab, office), or at the conclusion of a word (, malek, king). Moreover, synonymous terms (have the same meaning) are shown as several separate aspects. The Arabic text may be entirely distorted by negations; this should be taken into account. Therefore, Arabic SA requires additional research. Applying SA to English is less complicated (AlSalman, 2020; Al-Twairesh & Al-Negheimish, 2019). Moreover, Arabic Internet users mix Arabizi, a style of representing Arabic by English letters and numbers, Modern Standard Arabic (MSA), and multi-dialectical forms (Al-Azani & El-Alfy, 2020). This makes it more challenging and vague to judge Arabic thoughts (Farghaly & Shaalan, 2009). This intricacy therefore extends across the main stages of SA and Arabic text categorization, whether supervised or unsupervised. Al-Moslmi et al. (2018) stated that Arabic sentiment analysis (ASA) systems and resources are concerned about the rapid growth of elements created in localized Arabic dialects.

Stemming is one of the important steps for Arabic natural language processing (NLP). There are two groups in the Arabic stemmer algorithm. A light stemmer is applied initially to eliminate affixes (suffixes, infixes, and prefixes) and reveal the word’s root (Ghwanmeh et al., 2009). Snowball stemmer (Chelli, Aries & Mennoucho, 2018) and ARLStem (Abainia, Ouamour & Sayoud, 2017) are two illustrations of light stemmers. The second method is a heavy stemmer, which gets the root word by removing affixes and replacing some letters in words (Ghwanmeh et al., 2009). The Khoja stemmer (Khoja & Garside, 1999) and the Information Science Research Institute’s (ISRI) stemmer (Taghva, Elkhoury & Coombs, 2005) are a few examples of heavy stemmers, also known as root-extraction stemmers.

In NLP, word embedding outputs are required as input features in various models that use deep learning. Words in a vocabulary are converted into vectors of continuous real numbers (e.g., word hat → (…, 0.15,…, 0.23,…, 0.41,…S)) via the word embedding technology for modeling language and learning of features (Zhang, Wang & Liu, 2018). Typically, the method entails a mathematical embedding from a lower-dimensional dense vector space to a high-dimensional sparse vector space (such as one-hot encoding vector space, where each word occupies a dimension). Every dimension of the embedding vector corresponds to a word’s latent feature. Regularities and patterns in language may be conveyed by the vectors.

Fixed-length vector representations of words that encapsulate the semantics of generic phrases and linguistic patterns in natural language are known as pre-trained word embeddings (Asudani, Nagwani & Singh, 2023). Experts have put forth many approaches to get these representations. Many NLP applications benefit from word embeddings (Muaad et al., 2021).

Word embedding models fall into three categories: conventional, distributional, and contextual. Term frequency-inverse document frequency (TF-IDF), n-gram, and bag of words (BoW) models are the classifications for conventional word embedding. Probabilistic-distributional models, including word-to-vector (Word2Vec), global vector (GloVe), latent semantic analysis (LSA), latent Dirichlet allocation (LDA), and fastText model, are included in the distributional word embedding. The auto-regressive and auto-encoding contextual word embedding models include generative pre-training (GPT), bidirectional encoder representations from transformers (BERT) models, and embeddings from language models (ELMo).

This work’s primary objective is to close the gap left by the lack of a comparative empirical analysis of Arabic SA methodologies. Our goal is to clearly identify the best deep learning strategies for Arabic SA based on direct comparison using the same datasets. Furthermore, we desire to investigate the recently released data cleaning and natural language processing techniques that make use of the latest developments in Word Embedding models, such as fastText. In addition to the Arabic NLP processes of stemming, normalization, tokenization, and stop words, the framework provides a number of cleaning and preprocessing methods for ASA, such as Arabizi conversion, misspelling correction, emojis and dialects replacement.

This study seeks to respond to the following research questions: RQ1. What are the main issues with Arabic texts that can be resolved through the use of NLP and variant cleaning techniques for Arabic sentiment analysis?

RQ2. In comparison to the other sentiment analysis domains, how well can the deep learning and word embedding approaches be applied to the recently emergent food delivery services domain?

RQ3. How does the performance of the deep learning models for ASA change when different Arabic stemmers are combined with different word embedding techniques?

RQ4. How frequently may the deep learning classification process’ accuracy be increased by using a dataset with two balanced classes rather than three imbalanced classes?

This article looks at how the public felt about FDS during the outbreak in Arabic. It uses a variety of natural language processing (NLP) techniques along with deep learning algorithms to identify the emotions of the Arabic public. Using evaluations from Arabic about FDS, the system guides the reader through the process of assessing these emotions utilizing widely used deep learning techniques. Different levels of analysis were performed. We produced an Arabic dialect dictionary that includes the standard orientation of each dialect as well as other dialects. Using an already-existing emoji lexicon, emojis are extracted and identified. Several experiments were carried out using several deep learning classifiers with and without using pre-trained models, and using various word embedding and stemming methods. In this article, we examined four word-embedding methods: fastText (Bojanowski et al., 2017), GloVe (Pennington, Socher & Manning, 2014), and Word2Vec (Mikolov, 2013). Word2Vec uses two methods to create word vectors: Skip-gram (SG) and Continuous Bag-of-Words (CBOW). We evaluated four stemmers; ISRI, root-based Tashaphyne, stem-based Tashaphyne, and Snowball versus the non-stemmed technique. Furthermore, we conducted experiments on randomly initialized features’ weights, static and non-static weights, and features weighting based on custom embedding.

The rest of the article is organized as follows: In “Related Work”, pertinent issues and related studies focusing on Arabic sentiments are discussed. The system design and suggested approach are explained in “Proposed Methodology”. The examination of the experimental design and results is covered in “Experiments and Results”. The conclusion and the future work directions are outlined in “Conclusions and Future Work”.

Related work

Over the last few years, deep learning has been employed in SA for customers’ concerns for the Arabic language as well as other languages. However, there has not been much research on FDS, and to the best of our knowledge, only two earlier studies (Mustafa et al., 2023; Mustafa, Khabour & Shatnawi, 2024) focused on Arabic SA of FDS reviews. Meena & Kumar (2022) investigated FDS companies’ performance and customer expectations by exploiting social media data with machine learning. They found that in India, people are more focused on social accountability, whereas customers in the United States are more concerned with financial considerations.

In a study (Shaeeali, Mohamed & Mutalib, 2020), SA was performed on FDS customer evaluations on social media using four distinct AI algorithms (NLP, lexicon, text mining, and SVM). The highest accuracy rate was achieved by Lexicon (87.33%), which was followed by NLP, SVM, and then text mining, their accuracy rates were 71.67%, 69.70%, and 67.94%, respectively. Although various AI techniques have emerged, the most well-known ones are still SentiWordnet and TF IDF for lexicon-based approaches and Naive Bayes and SVM for ML approaches (Drus & Khalid, 2019). Although they are more accurate than ML models, syntax-based approaches are difficult to implement for SA in languages other than English.

The impact of traffic conditions on food delivery services’ major performance indicators (which include customer reviews and the physical locations of restaurants provided by Facebook) was examined by Correa et al. (2019) using the Google Maps API. Salminen et al. (2022) research showed that while human raters perform significantly worse than the assessed methods, a machine classifier can identify false reviews almost perfectly. The results suggest that, whereas humans find it challenging to identify fraudulent reviews, “machines can battle machines” in this regard. Their conclusions have ramifications for review platform accountability, consumer protection, and business defense against unfair competition.

Beside texts classification, image-based classification is another mechanism for sentiment analysis. By uploading and sharing photographs to different social media platforms, users can express their thoughts, feelings, and views. Transfer learning-based models have shown remarkable advances in picture classification in recent years (Karray et al., 2008). Transfer learning techniques enable the effective sentiment analysis of images. In order to classify image sentiment, Meena & Mohbey (2023) compared and examined a number of transfer learning approaches. Using popular image sentiment datasets including CK+, FER2013, and JAFFE, the results are examined and contrasted. When evaluating the results, a number of measures are taken into account, such as recall, accuracy, and precision. Using the CK+ dataset, they attain the highest image sentiment analysis accuracy of 99.57% with the Inception-v3 model.

One of the most emerging techniques for creating a point of interest (POI) recommendation system is sentiment analysis. Two models are combined for POI recommendation in the study by Meena et al. (2024). The first model is trained on an election dataset and expects attitudes using bidirectional long short-term memory (BiLSTM). The accuracy of the recommended approach is found to be higher than that of the current models (99.52%). The class labels are then predicted using this model on the Foursquare dataset. The creation of user and POI embeddings comes next. Using the LSTM model, the following model suggests to the user the best points of interest and associated coordinates. POIs from the Foursquare dataset are suggested based on user interest and location filters. The suggested system can be used to advise interesting places to certain users, as well as to advise trips and groups.

Deep long short-term memory (LSTM) models were used by researchers Imran et al. (2020) to estimate sentiment polarity, providing intriguing insights into social media users’ collective responses to the coronavirus outbreak. Nilashi et al. (2022) combined machine learning and survey-based approaches for customer satisfaction analysis during the COVID-19 outbreak. Wang et al. (2020) examined public concerns by selecting Weibo posts related to COVID-19 to identify negative sentiment characteristics.

Convolutional neural network (CNN) and recurrent neural network (RNN) (Elman, 1990) are the primary DL architectures used to manage NLP functions (Yin et al., 2017; Goldberg, 2017). The NLP challenge determines these DL structures’ performance and results (Yin et al., 2017). Gated recurrent units (GRU) (Cho et al., 2014) and LSTM networks were created to address several drawbacks discovered in RNN. Compared to ordinary RNN and CNN, LSTM-RNN networks, with their gated RNN design, are superior at remembering longer sequences for text categorization (Tang, Qin & Liu, 2015; LeCun, Bengio & Hinton, 2015). According to the latest findings, an LSTM network can handle most NLP functions fairly well (Chiu & Nichols, 2016; Yin et al., 2017). Text classification relies heavily on the textual extraction of features, which has a direct impact on the text’s classification accuracy (Liang et al., 2017). Language data must be provided in a meaningful manner for additional processing because it is a series of distinct symbols. Features for characters, singular phrases, sentences, words, or even documents may be developed based on the NLP difficulty (Goldberg, 2017). Techniques for extracting text features include fusion, mapping, clustering, and selection (Liang et al., 2017). In addition to classification, DL techniques have been actively employed increasingly to construct distributed feature representations of words or even characters. Text can be vectorized into numerical values in high-dimensional space using neural networks. This technique, known as embedding, can be used with words, n-grams, or characters. One helpful technique for obtaining numerical features from words is word embedding (WE). Every word appears as a vector in high-dimensional space in WE. Through the spatial mapping of adjacent and related words, this is a useful technique for preserving word relationships. The three most prominent embedding methods: Word2Vec (Mikolov, 2013), GloVe (Pennington, Socher & Manning, 2014), and FastText (Joulin et al., 2016) have all been utilized extensively for stable and precise WEs. However, gathering a sizable text corpus necessitates time and computer resources when training these WEs from scratch. Alternatively, a range of publicly available pre-trained WE vectors can be accessed for NLP demands. The Arabic WE methods are offered by a few pre-trained WEs.

Bojanowski et al. (2017) published 249 pre-trained word vectors, two of which were for the Arabic language, using the FastText approach. Multiple variants of AraVec (Soliman, Eissa & El-Beltagy, 2017) are based on the Word2Vec model’s construction techniques, namely skip-gram and continuous bag of words (CBOW) techniques. 3.3 billion words, covering MSA and Egyptian dialects, were used in its training. ArabicNews (Altowayan & Tao, 2016) used 190 million terms from MSA corpora gathered from various sources to create pre-trained WE vectors using the Word2Vec technique. Alwehaibi & Roy (2018) research compares the effectiveness of different Arabic pre-trained WE vectors for classification. The classification model employed was the LSTM-RNN. After comparing three different WE approaches for sentiment classification, they found that the FastText pre-trained embedding vector was the most effective. On the training and testing subsets, it achieved an accuracy value greater than 90%.

Another method for recognizing text at the character level as opposed to the word level is character embedding (CE) (Zhang, Zhao & LeCun, 2015). The dialectical words might not have any embedding vectors and become unrecognized because the majority of pre-trained WE vectors depended on MSA. Since every word in MSA and dialects uses the same group of characters, using characters instead of words might solve the problem of unknown words when extracting text features. The design of CNN is important for character-level embedding construction. Other studies have employed the RNN-LSTM model (Kim et al., 2016) for word-level embedding, or very deep CNN (Conneau et al., 2016) to increase the performance.

Alayba & Palade (2022) used an accessible big Arabic collection that comprises over 1.5 billion words named the Abu El-Khair Corpus (El-Khair, 2016) for word embedding. Just three word-embedding strategies were taken into consideration: Word2Vec, GloVe, and fastText. Word2Vec uses two methods to create term vectors: Skip-gram (SG) and CBOW.

Comparing Arabic SA with other languages reveals that relatively few studies have looked at the language (Guellil, Azouaou & Mendoza, 2019; Khabour, Al-Radaideh & Mustafa, 2022; Abo, Raj & Qazi, 2019). Local dialects were disregarded in favor of MSA in earlier ASA research. MSA is more conceptually rich and complex than English, although it is more parse-friendly than Arabic dialects, which are primarily used in news. While MSA follows standardization, dialects lack it and contain a multilingual component (Abdul-Mageed, Diab & Kübler, 2014). Older ASA systems failed because languages were so common on social media (Aldayel & Azmi, 2016). Some research has attempted to improve the Arabic sentiment sources by utilizing a bilingual methodology based on the popularity of English sentiment sources.

That being said, this study differs from others in that it presents a novel, all-encompassing model that uses DL and NLP techniques to investigate Arabic attitudes toward FDS. A CNN, BiLSTM, and concatenated LSTM-CNN are created as part of the state-of-the-art model for Arabic sentiment categorization presented in this study. Additionally, we looked into the four ways of random weights, pre-trained static and non-static weights, and custom weights, for feature representation and weighting. We tested four alternative vector dimensions for the random-weights-based method: 50, 64, 100, and 300. We pre-processed the Arabic words using a variety of methods, including different stemmers. On the same corpora, we compared Word2Vec (CBOW and skip-gram), GloVe, and fastText in the pre-trained word embedding situation. Semantic and morphological analysis are among the many NLP techniques used. An Arabic dialect lexicon was produced. We created a brand-new method for identifying and swapping out dialects and emojis.

Proposed methodology

In our earlier work (Mustafa et al., 2023; Mustafa, Khabour & Shatnawi, 2024), we suggested an approach that combines features selection with a machine learning model to perform sentiment classification on FDS reviews for Arabic text. However, in this article, we used a variety of strategies to create the model to enhance sentiment categorization. In the input layer, various Arabic stemming algorithms and word embedding strategies were applied. The CNN, BiLSTM, and Hybrid LSTM-CNN deep learning classifier models are among the many that we produced. To gain an understanding of the proposed method, Fig. 1 summarizes its overall phase’s pipeline.

Figure 1 The overall flowchart of the undertaken methodology.

Reviews dataset description

The reviews dataset was introduced by Mustafa et al. (2023). The primary information source we used to gather reviews specific to FDS was the FDS Talabat. Talabat takes online orders from different eateries and offers delivery services to patrons. Customers can review restaurants, drivers, and the app itself through the app’s rating features. Customers can also share their experiences by posting reviews online or by filing complaints. These reviews were gathered both manually and with the aid of an online scraper tool. After that, native Arabic speakers personally labeled each review as positive, negative, or neutral. Only “Arabic” reviews are included in the data collection process at this time. The neutral reviews are very low in the dataset, and this may affect the performance of the classifier towards this class. Figure 2 illustrates the unbalanced multiclass dataset under study. Since Talabat operates throughout a number of Middle Eastern nations, including Jordan, Iraq, and Egypt, as well as Gulf nations like Saudi Arabia and Bahrain, the reviews include the viewpoints of individuals from various backgrounds and dialects, so this will pose a challenge in these reviews.

Figure 2 Classes of reviews in the FDS dataset: positive (1), negative (−1) and neutral (0).

Additionally, as shown in Fig. 3, the most commonly used terms related to FDS were examined and represented as word clouds, where each word’s size indicates its significance in the text. Figure 3A includes words that appeared in the negative reviews without any preprocessing steps. Figure 3B includes words that appeared in negative reviews after the preprocessing steps were applied. Similarly, Fig. 3C includes words that appeared in the positive reviews without any preprocessing steps. All the words in the positive reviews after applying the preprocessing steps are appeared in Fig. 3D.

Figure 3 Word clouds based on words that appeared in the (A) Negative reviews without any preprocessing steps.

(B) Negative reviews after the preprocessing steps. (C) Positive reviews without any preprocessing steps. (D) Positive reviews after applying the preprocessing steps.

Dataset cleaning and NLP steps

Numerous pointless and repetitious reviews are present in the raw datasets, which ought to be effectively removed. Furthermore, people frequently use emojis to express their feelings and sentiments. The cleaning phase includes various automatic and manual steps. The automated multistage cleaning process includes removing duplicated reviews and replacing emojis with appropriate Arabic text that reflects their sentiments (positive, negative, neutral) based on its higher score in Kralj Novak et al. (2015) dictionary.

Arabic language evaluations would contain “Arabizi,” in which Arabic words are written in Latin script (Baly et al., 2017). Franco terms—things like “”, which are represented as “mat3m” are Arabic words written with a combination of English letters and numbers—have thus been swapped with their Arabic synonyms. Since Arabizi reviews and comments are usually considered spam, SA scholars usually remove them from the dataset. We have adopted a different approach for this task: we have developed a rule-based converter that can change reviews from Arabizi to Arabic. The rule-based converter first transforms a review into a list of words or tokens, using punctuation and white spaces as separators. Second, the Arabizi letters are converted to their associated Arabic alternatives using the translations given in Fig. 4 to convert each Arabizi word into an Arabic word, followed by replacing vague words and phrases with clear formal text utilizing our dialects dictionary that was built in the prior work (Mustafa, Khabour & Shatnawi, 2024), and extended in this work by adding more dialects. After handling Arabizi text replacement, the last step is to correct misspellings. Annotating reviews for testing as positive, negative, or neutral is done manually by native Arabic speakers.

Figure 4 Arabic characters and their corresponding translations in Arabizi.

Following cleaning, several NLP techniques are applied, starting with normalization, where we apply various normalization strategies. First of all, characters with various forms are converted to a single form, such as the various forms of “alif” as . Dropping diacritics, punctuation, special characters, and digits, as well as stripping tatweel and repeated characters, are also applied. Moreover, the Arabic definite article (the) was eliminated. We implemented the “Removing stopwords” NLP technique utilizing the stopword list of CLTK. We have extended this list to include more missing stopwords. Tokenization of sentences and words is utilized to split the text into separated units. The pyArabic library contains the majority of Arabic normalization approaches (Zerrouki, 2023).

Finally, stemming is used to reduce the text size in a variety of ways. In this stage, we have evaluated the employment of three Arabic stemmers: ISRI, Tashaphyne, and Snowball. ISRI stemmer, developed by Taghva, Elkhoury & Coombs (2005), has put into practice an Arabic root-extraction stemmer that is comparable to the Khoja stemmer but does not have a root dictionary. In addition, if no root was found, the ISRI stemmer returned the normalized form rather than the original, unaltered word. Tashaphyne, supported in Python, provides both stemming and root extraction, unlike the Khoja, ISRI, and Assem. The Snowball stemmer algorithm, also known as the Porter2 stemming algorithm, is an enhanced version of the Porter Stemmer that addresses some of its shortcomings (Porter, 2001). Snowball supports several languages, including Arabic. The Arabic Light Stemmer by Assem is a snowball-based Arabic stemming algorithm (Chelli, Aries & Mennoucho, 2018). It offers only basic text normalization and stemming.

One of the objectives of this study is to measure the performance impact of different stemming strategies, such as light stemmers and root-extraction stemmers. Table 1 provides an example demonstrating the difference between the stemmers based on their strategies to remove appendages such as prefixes and suffixes.

Table 1 Comparison between Arabic text-stemming techniques.

Text-stemming technique	Example (Arabic)	
Original FDS review text		
Cleaned Review		
Preprocessed review without stemming		
Preprocessed review with root stemming by (ISRI)		
Preprocessed review with root stemming by (Tashaphyne)		
Preprocessed review with light stemming by (Tashaphyne)		
Preprocessed review with light stemming by (Snowball)		

Features representation and word embedding methods

Word embeddings are vector representations of terms that are distributed; terms with equivalent meanings have similar vector representations. The cosine similarity of the words could be used to calculate their separation. The terms “traveling” and “vacation,” for instance, will be represented by vectors that are closer together. Compared to bigram and trigram techniques, which either ignore or impose associations between phrases or tokens, this is a more descriptive method for conveying text than classic methods like the bag-of-words. Dense vectors make it simpler for deep learning models to converge than sparse ones. Furthermore, the nature of the dataset and the business need are always determining factors. The embedding layer is a dictionary that links integer indices to dense vectors with real values rather than only 0’s and 1’s. The words are dispersed throughout a very high dimensional space, while the vector’s size is fixed. The model receives as input a very big corpus of text, and the words are dispersed into vectors based on that. A variety of methods exist for word embeddings, including GloVe (Pennington, Socher & Manning, 2014), Word2Vec (Mikolov, 2013), and fastText (Bojanowski et al., 2017). Different parameters are taken into account by each technique for representing the words. The input row of the text is an important function in word representation, where word distribution can be altered by word context. In this article, we only looked at three word-embedding methods: fastText (Bojanowski et al., 2017), GloVe (Pennington, Socher & Manning, 2014), and Word2Vec (Mikolov, 2013).

Word2Vec model

Mikolov (2013) developed the Word2Vec technique. Word2Vec is a neural network prediction model that learns word embeddings from text, thereby making it computationally effective. The approach considers every word in a big corpus that is provided as input. The premise behind this approach is that words occur in comparable situations and have similar meanings (Harris, 1954). Using a fixed-size window, the method changes the word vectors based on where the word appears in the immediate context. These terms have greater similarities, and the vectors will converge. It contains the CBOW and the Skip-Gram models (SG) (Mikolov, 2013). Using its context words , the CBOW model predicts the particular word (e.g., ), whereas the SG model does the opposite, forecasting the context words given the particular word. By considering the whole context as a single observation, the CBOW model statistically smoothes over a significant amount of distributional information. It works well with smaller amounts of data. However, the SG model, which performs better with larger amounts of data, views every context-target pair as a novel observation. At time step t, the target’s CBOW is the word wt. The model is given a window of n words surrounding wt, and the loss function J can be expressed as in the Eq. (1) (Do et al., 2019):

(1) J=1T∑t=1TlogP(wt∣wt−n,…,wt−1,wt+1,…,wt+n).

On the other hand, the skip-gram model predicts the neighboring words wt+j using the center word wt. The objective function in this instance is as in Eq. (2) (Do et al., 2019):

(2) J=1T∑t=1T∑−n≤j≤n,n≠0tlogP(wt+j∣wt).

GloVe model

Another word embedding method that was put forth by Pennington, Socher & Manning (2014) is called GloVe. In this method, word embedding vectors in a space are constructed by an unsupervised learning technique. This model’s objectives of grouping related words and avoiding dissimilar words are comparable to those of Word2Vec. Nevertheless, this method’s methodology stands apart from Word2Vec’s. GloVe analyses every word in the corpus in addition to taking into account the word’s context, which consists of the words that are located around it. In order to distribute the word vectors in this model, both local and global parameters from the corpus are required. The non-zero values in a global word-to-word co-occurrences matrix are the primary concern of this approach. It calculates the ratio of these two words’ joint co-occurrence probability in the input data. If the ratio is high, the affinity of these terms can be stated, and vice versa.

By assuming the frequency with which the terms wi and wj co-occur within a specific context window is indicated by each element Xij in the matrix. Xi represents how many times a specific word occurs in relation to the term i. According to Eqs. (3) and (4), the Pij is the probability that the word j will occur in the context of the word i (Asudani, Nagwani & Singh, 2023):

(3) Xi=∑kXik

(4) Pij=P(j∣i)=XijXi.

The link between a co-occurrence matrix and a word embedding is roughly represented by a weighted least squares regression model (Asudani, Nagwani & Singh, 2023). A weighting function for the vocabulary of size V is given by the function f(Xij). The context word vectors are denoted by w~, while the word vectors are represented by w. To restore the symmetry, the terms bi and bj~ are biased for the words wi and wj. As seen in Eqs. (5) and (6), a weight function f(x) makes sure that the weight is not substantially raised when the term’s frequency is excessive.

(5) J=∑i,j=1Vf(Xij)(wiTwj~+bi+bj~−log⁡Xij)2

(6) f(X)={(xxmax)34ifx≤Xmax1otherwise..

fastText model

Another word embedding method known as fastText was introduced by Bojanowski et al. (2017). The core of it is an unsupervised technique that uses vectors for denoting words. It is an expansion of the Word2Vec framework in which the sub-words are taken into account. It also offers two architectural models: SG and CBOW. But each word in fastText is separated into an n-gram character. Angle brackets are used as a unique boundary to show where a word starts and ends. It is used to distinguish a word from its subword and the word itself. The word “sentiment,” for example, has the fastText representation (sen, sent, enti, ntim, time, ment, ent) when n = 4. The terms sent and time are denoted by the sequences (sent) and (time); they are not the same as the 4-gram sent and time from the original term sentiment. This model can distinguish between a word’s shorter character sequences and prefixes and suffixes by taking sub-words into account. The word itself will be represented in a vector with its set of character n-grams in this model. A hashable list is used to link the sub-words to their parent word, and the vector of the primary word is equal to the sum of the n-gram vectors. FastText works well with infrequent words. Consequently, by breaking a word up into n-grams, its embeddings can still be recovered even if it was not detected during training. In Word2vec or GloVe, respectively, words that are not included in the model vocabulary have no vector representation. This strategy has this important advantage.

A word set of varying lengths makes up each FDS review. To work with integers, a word-to-index dictionary should be constructed. In the word-to-index dictionaries, every term in the corpus functions as a key, and the value of each key is determined by its associated unique index. In this stage, each word in the review is replaced by its corresponding word embedding, which can be either one of two types: pre-trained distributed word representations or one that is specifically built for the review. BOW, Skip-Gram, fastText, and GloVe are the four different embeddings that we use. In this study, the AraVec 3.0 (Soliman, Eissa & El-Beltagy, 2017) project was used to produce unigram models of the Arabic Skip-gram and CBOW models. Both models were developed using tweets that were taken directly from Twitter. Each model contains 1,259,756 vocabularies, and each model’s vector size is 300. The corpus employed in this study’s Arabic GloVe model is made up of 1.75 billion tokens, 1.5 million vocabulary words, and 256 vector sizes. The Arabic language’s standardized fastText word vectors were pre-trained using the Common Crawl and Wikipedia resources. For the training, a position weight with a dimension of 300 and a continuous bag of words (CBOW) was employed. According to the fastText library (Facebook, 2024), the character n-grams have a length of five characters. The details of the chosen CBOW, Skip-Gram, GloVe and fastText models are provided in Table 2.

Table 2 Description of the examined pre-trained Arabic words embedding.

Pre-trained word embedding model	Model corpus domain	Number of documents	Number of vocabularies	Vector dimension	
CBOW	Twitter (unigrams)	66,900.000	1,259,756	300	
Skip-Gram	Twitter (unigrams)	66,900.000	1,259,756	300	
GloVe	Multiple resources such as Arabic tweets and Wikipedia	————-	1,500,000 (1.75 billion) tokens	256	
FastText	Common crawl (Wikipedia)	—————	——————–	300	

Each text review is displayed in the aforementioned model as a 2D vector with a dimension of n × d, where n is the term count in the text review and d is the vector representation of a word’s dimension length. We have assessed different vector dimension lengths, including d = 50, 64, 100, 256, and 300. We adopt the padding strategy, to pad the representation of each review with zeros, to guarantee that all reviews are the same fixed size. In this manner, each review will have a size of n′ × d, where n′ was set at 100 for all experiments. Table 3 demonstrates how to convert each FDS review to a vector representation using word embedding.

Table 3 Example of converting textual review to a vector representation using word embedding.

Text-stemming step	Example (Arabic)	
Origin review text		
Tokenized review after cleaning, preprocessing, and stemming		
Text indexing (encoding)	[2, 3, 14, 161, 304, 70, 43, 40, 1, 82, 71, 63, 3,148]	
Padding	[2, 3, 14, 161, 304, 70, 43, 40, 1, 82, 71, 63, 3,148,….., 0,0,0,0,0,0,0,0,0,0]	
Vector representation using word embedding	[[5.222600e−02, 4.599880e−01, −2.192200e−01, −1.810400e−02, 4.585380e−01, 2.424030e−01, 1.037320e−01, 4.975420e−01, −8.883430e−01, −4.546760e−01, 8.732000e−02, 3.694500e−02,…], [−3.264180e−01, −2.968420e−01, 5.262500e−02, 3.534710e−01, −7.600600e−01, −6.230630e−01, 6.521930e−01, 3.025940e−01,…], [−1.751270e−01, 2.583510e−01, −2.118470e−01, 4.183250e−01, 1.825560e−01, −3.380570e−01, −3.754840e−01, −5.319610e−01,…], …….. ]]	

Sentiment classification and deep learning models

Deep learning has been frequently employed by authors in the field of SA, where it demonstrated encouraging outcomes and effective performance (Ain et al., 2017). Using massive datasets, deep learning methodology is applied to build high-level model abstractions and nonlinear transformations. Typically, deep learning text categorization model architectures are made up of the following elements connected in a particular order as depicted in Fig. 5:

- Embedding layer: keeps track of a lookup table that transforms words with numerical indexes into their dense vector representations.

- Deep network: The deep network receives the embedding vectors and transforms them into a compressed representation. The compressed representation fully captures the information included in the text’s word sequences. RNNs or variations of them, such as LSTM/GRU, typically make up the deep neural network component. To combat the propensity for overfitting, the dropout layer is added.

- Fully connected layer: converts the RNN/LSTM/GRU’s deep representation into the final output classes or class scores. This component is made up of fully linked layers, batch normalization, and optional dropout layers for regularization.

- Output layer: this layer produces either Sigmoid for binary classification or Softmax for both binary or multi-classification, depending on the problem at hand.

Figure 5 General deep learning architecture for sentiment analysis.

Our experimentations considered four groups to test sentiment classification: (1) using WE which is learned from scratch. (2) Using static pre-trained WE. (3) Using non-static pre-trained WE, and (4) using specially generated WE. Furthermore, experiments utilized three DNN topologies: CNN, BiLSTM, and a hybrid LSTM-CNN.

Convolutional neural network model

CNN has been successfully used to solve challenges involving document classification and natural language processing (Albawi, Mohammed & Al-Zawi, 2017). The proposed CNN model has an embedding layer with the following settings: embedding-size = 50, 64, 100, 256, and 300 (depends on the used WE method)

max-len set to “100” and max-features set to num-unique-word (depends on the used stemmer)

5-kernel, rectified linear (or “relu”) activation function is utilized with 200 filters (parallel fields for word processing) in a conservative CNN configuration dropout = 0.3

MaxPooling1D with pool-size = 5. The pooling layers can be viewed as a method of downsampling (decreasing the size of) the input feature vectors. The dropout layer is one of the most effective and commonly used regularization techniques in NNs.

Then, the model utilized another convolution layer with 200 filters, having a 5-kernel, rectified linear (or “relu”) activation function. GlobalMaxPooling1D layer is added here. Whereas MaxPooling1D or GlobalMaxPolling1D layers’ primary responsibility is to lessen the dimensionality of each convolutional output, Conv1D layers are responsible for computing the convolution operations. The MaxPooling window then outputs patches of maximum values by extracting the largest value contained inside following the convolution operation. A 32-dense layer with the sigmoid activation function is used to empower the network’s ability to classify the extracted features better, followed by another dropout layer of 0.3. The output layer outputs a value of 0, 1, or 2 for the review’s neutral, positive, and negative feelings, respectively, using a softmax activation function. A dense layer with three output units representing the three potential classes of our dataset is being stacked. ’Softmax’ is an excellent activation function to use in the final layer if producing probabilistic outputs. Figure 6 shows the overall CNN model layers architecture in the specified order.

Figure 6 CNN model layers architecture.

BiLSTM model

Hochreiter (1997) describes LSTM as a unique type of RNN that can recognize long-term dependencies. The long-term dependency issue that the conventional RNN suffered is specifically avoided by LSTM architecture. These networks are used for recurring objects, such as time series data. They can understand the connections between the components of an input sequence. In numerous text mining and English SA projects, they showed excellent accuracy (Tai, Socher & Manning, 2015). Unlike the LSTM, which reads the sequence from left to right, the bidirectional LSTM (Bi-LSTM) combines two layers that are moving in contrary directions (forward and backward) over the same output. The output layer concurrently accepts data from the prior sequence (backward) and subsequent sequence (ahead) states. As a result, it is very helpful in situations when the context of the input is needed, for instance when the negation phrase comes after a positive word (Schmidhuber, 2015). Our BiLSTM model includes an embedding layer with the following settings: embedding-size = 50, 64, 100, 256, and 300 (depends on the used WE method), max-len set to “100” and max-features set to num-unique-word (depends on the used stemmer). The following layer includes 64 units of BiLSTM layer followed by a fully connected sigmoid activation layer with 32 output units and a 0.3 dropout layer. Ultimately, a dense softmax layer that forecasts the three sentiment class labels is combined. Figure 7 shows the overall BiLSTM model layers architecture in the specified order.

Figure 7 BiLSTM model layers architecture.

LSTM-CNN model

We proposed a hybrid LSTM-CNN model that combines many layers together for better sentiment classification results. This model includes an embedding layer with similar settings as in the CNN and BiLSTM models: embedding-size = 50, 64, 100, 256, and 300 (depending on the used WE method), max-len set to “100” and max-features set to num-unique-word (depends on the used stemmer). The next stage consists of two LSTM layers with 200 and 64 output units respectively, followed by a convolutional layer with 200 filters, and a kernel of size 5. A Rectified Linear Unit (ReLU) activation function is used due to its effectiveness and low computational complexity. To downsample feature maps produced by the convolutional layers, MaxPooling1D layer with a pool-size of 5 is added along with a dropout layer set to 0.3. The same three previous consecutive layers are repeated (the Conv1D, MaxPooling1D, and dropout) with the same settings. We applied a 32-dense layer with sigmoid activation to a fully connected layer after flattening the results into a single vector. Finally, a 3-dense with a softmax function that forecasts the three labels is combined. Figure 8 summarizes the overall LSTM-CNN model layers architecture in the specified order.

Figure 8 LSTM-CNN model layers architecture.

Experiments and results

Experiments setup

The Talabat online food delivery service (FDS) is the subject of the experimentation dataset, which is linked to a raised concern in the COVID-19 emerging catastrophe. The original collection and use of the dataset were made by Mustafa et al. (2023). The users decided to share their thoughts on this matter on the Talabat website and application. The web scraper program was used to gather the dataset between October 1, 2021, and December 3, 2021. Table 4 contains information about the dataset statistics.

Table 4 General description of Arabic FDS reviews dataset.

Key statistic	Total	
Whole reviews	30,948	
Distinguished reviews	30,799	
Positive reviews	17,419	
Negative reviews	10,632	
Neutral reviews	2,748	
Whole number of words	286,247	
Distinguished words	44,804	
Average words in a single review	9.18	
Average characters in a single review	49.75	

The three deep learning models will be used in the experimental setup to create and evaluate models in a Python 3.7.6 environment for text review classification on identical hardware and operating systems. All experiments have the following configurations: - Operating system: Windows 11 Education 64-bit (10.0, Build 18363)

- System manufacture model: HP Inc. Precision 5720 AIO

- Processor: Intel(R) Xeon(R) CPU E3-1275 v6 @ 3.80 GHz (8 CPUs), ~3.8 GHz

- Memory: 16,384 MB RAM (16 GB)

- GPU: Radeon (TM) Pro WX 7100 Graphics.

Table 5 lists several library functions that were used.

Table 5 Installed Python packages.

Package name	Version	
Keras	2.11.0	
Tensorflow	2.11.0	
Pandas	1.3.5	
Sklearn	0.21.3	
Nltk	3.4.5	
Cltk	0.1.113	
Re	2.2.1	
Numpy	1.21.6	
Gensim	4.2.0	

The parameters for embedding can be either specified as the length of the vector, while others can be trained as weights. In the training phase, weights are learned by the model and for the dense layer too. Related groups of words that are closer together can be found in the corresponding embedded vectors (Al-Bayati, Al-Araji & Ameen, 2020). Word embeddings can be learned using a variety of techniques, including Word2vec (Google), GloVe (Stanford), FastText (Facebook), and the Embedding layer (Keras). The embedding layer is the initial hidden layer of the model that prepares data to be fed as input for the subsequent layer of the model. In this work, every one of the four learning techniques has been tested and used.

The Sequential model is a stack of linear layers, it offers the ability to utilize any of the many Keras-compatible layers. The Dense layer is the standard densely linked neural network layer with all the weights and biases. The Sequential model can be constructed by adding layers one at a time. Additionally, including a list of indicators that can be utilized for future evaluation but have no impact on training. In this case, we want to employ the Adaptive Moment Estimation (Adam) optimizer (Kingma & Ba, 2015) along with the categorical cross entropy as the loss function, which is suitable for multi-class classification. The dataset is split into two parts: 90% for training, and 10% for testing. During the training process, 10% of the training data were utilized as validation data. Additionally, we performed five-fold cross-validation by averaging the results of five distinct runs for each data division, which yields enhanced performance. After training is completed, the embedding layer learns the weights, which are simply the vector representations of each word. The goal of separating testing from the validation is to select the model with the highest validation accuracy and then test it with the testing set. We must define three parameters for the input layer: the vocabulary size, the embedding dimension, and the sequence length. As the stemmer algorithm changes, so does the vocabulary size. Table 6 displays the various vocabulary sizes for each stemmer employed. Table 6 estimates how many of each vocabulary term was discovered in each pre-trained WE model. The no-stemming technique provided the bigger vocabulary for the FDS dataset, as it includes (28,035) words, whereas Tashaphyne’s root stemming strategy generates the least vocabulary, as it comprises (7,420) words. Also, utilizing GloVe WE always delivers greater coverage across all embedding approaches.

Table 6 The number of unique words using each stemming technique and word embedding coverage.

Stemming	Stemmer	Vocabulary	BOW/SG	FastText	GLoVe	
technique		Size	coverage	coverage	coverage	
No stemmer	————	28,035	18,260 (65.1%)	20,926 (74.6%)	21,124 (75.3%)	
Root stemming	ISRI	11,418	6,667 (58.4%)	7,253 (63.5%)	7,438 (65.1%)	
	Tashaphyne	7,420	4,672 (63%)	4,410 (59.4%)	4,780 (64.4%)	
Light stemming	Tashaphyne	14,482	9,519 (65.7%)	9,587 (66.2%)	9,813 (67.7%)	
	Snowball	17,618	12,639 (71.7%)	12,533 (71.1%)	12,878 (73%)	

It is necessary to provide the number of iterations that the model will go through during training. The completed iterations are commonly referred to as epochs. For all experiments, the epochs are initially set to 20. The Early Stopping observed validation loss during the training process, which was used to avoid over-fitting and automatically alter the number of epochs. If the Early Stopping determined that the validation loss rose for three iterations (patience = 3) during training, the training model would be terminated. The batch size determines how many samples are utilized in one epoch or how many samples are used in one forward/backward pass. We changed the size to 16, 32, and 64 each time. The batch size 64 gives us the greatest results in our trials by tracking accuracy and loss, which can suggest overfitting when the training accuracy increases and the validation accuracy decreases. It might be argued that the large batch size has a negative impact on validation accuracy, convergence speed, and execution time. Many researchers recommend using smaller batch sizes (fewer than 64) to improve performance. The learning rate was set to 0.001. This rate governs the mechanism for updating the weights when each batch is completed. After conducting numerous trials with different hyperparameters using the trial-and-error method, the hyperparameters utilized in the basic models were chosen. To determine the ideal hyperparameter values, exhaustive experiments were carried out; Table 7 displays the hyperparameters that produced the best results for the model.

Table 7 Hyperparameters’ settings for all deep learning models.

Layer	Models used the layer	Hyper-parameters’ settings	
Embedding	CNN, BiLSTM, LSTM-CNN	Embedding size: 50, 64,100, 256, 300 Max length: 100	
Conv1D (1)	CNN, LSTM-CNN	Kernel size: 5 Number of filters: 200 Activation: Relu	
Conv1D (2)	CNN, LSTM-CNN	Kernel size: 5 Number of filters: 200 Activation: Relu	
Dropout (1)	CNN-BiLSTM, LSTM-CNN	Dropout rate: 0.3	
Dropout (2)	CNN, LSTM-CNN	Dropout rate: 0.3	
MaxPooling1D (1)	CNN, LSTM-CNN	Pool size: 5	
MaxPooling1D (2)	LSTM-CNN	Pool size: 5	
GlobalMaxPooling1D	CNN	——————-	
Bidirectional Lstm	BiLSTM	Units: 64	
Lstm (1)	LSTM-CNN	Lstm units: 200	
Lstm (2)	LSTM-CNN	Lstm units: 64	
Flatten	LSTM-CNN	——————-	
Dense (1)	CNN, BiLSTM, LSTM-CNN	Dense units: 32 Activation: Sigmoid	
Dense (2)	CNN, BiLSTM, LSTM-CNN	Dense units: 3 Activation: Softmax	
————-	CNN, BiLSTM, LSTM-CNN	Learning rate: 0.001 Number of epochs:20 Batch size: 64	
		Optimizer: Adam Loss: Categorical Cross entropy	
		Data split: 90% training–validation, 10% testing	

Evaluation metrics

Recall (R) indicates a classifier’s completeness, while precision (P) evaluates a classifier’s exactness. P and R can be combined to provide the single metric known as F1, sometimes known as the F-score, which is the weighted harmonic mean of both measures. The weighted accuracy is calculated as well to evaluate our model. The calculations of R and P are shown in Eqs. (7) and (8):

(7) R=TPTP+FN

(8) P=TPTP+FP.

True positive (TP) defines how often the model categorizes a positive sample as positive correctly. False negative (FN) defines how often the model misclassifies positive samples as negative. False positive (FP) defines how often the model misclassifies a negative sample as a positive one. True negative (TN) defines how often a negative sample gets the right classification from the model. For multi-class classification, we want to find TP, FN, FP, and TN for each class separately. For example, in Fig. 9 which presents a 3-class confusion matrix, let’s designate the target to be the C1 class. All other classes have negative classifications, while this one has a positive classification. Then Fig. 9 demonstrates how to determine each of them for class C1.

Figure 9 3-class confusion matrix.

The difference between the issued real value and the model’s prediction is referred to as loss. The greater the loss, the less accurate the model. A categorical cross-entropy function is used to determine the loss for multi-class classification. As a result, loss evaluation is depicted in Eq. (9) (Grandini, Bagli & Visani, 2020).

(9) Loss=−∑i=1syi×log⁡yi^

where yi is the goal value, yi^ is the ith scalar value in the model output, and s is the number of scalar values in the model output.

Results and discussion

We carried out extensive research to evaluate the various elements for deals with word embeddings such as the effect of using the various stemmers to create the FDS reviews corpus vocabulary, the various weighting vectors methods, random, static, non-static, and custom, and the effect of embedding vectors dimension size. Each method has its advantages and disadvantages over the other.

Experiments (A): learning embedding from scratch

During the first group experiments, we examined learning a word embedding as part of fitting a neural network model, where random weights are used to initialize the word vectors, and then they are updated during the training process using a backpropagation method. We also evaluate the impact of various stemming techniques across various embedding dimension sizes. The results of ASA employing random weight initialization approaches in conjunction with deep learning methods are displayed in Fig. 10. This includes CNN-rand, BiLSTM-rand, and LSTM-CNN-rand results from 3-Class Sentiment Classification Using Deep Learning. We started from scratch when training the embedding layer, using four different stemmers (ISRI, stem-based Tashaphyne, root-based Tashaphyne, and Snowball) and dimension sizes (50, 64, 100, and 300).

Figure 10 Results of 3-class sentiment classification using deep learning (CNN-rand, BiLSTM-rand, LSTM-CNN-rand).

Learning embedding from scratch is generally not particularly precise when dealing with sequential data. Techniques that take into account local and sequential data rather than absolute positional data are the focus when working with sequential data. This is demonstrable through performance. Using Tashaphyne-based roots and the BiLSTM classifier model, the accuracy results achieve the highest value of 83.63% when the embedding dimension is adjusted to 300, as per the scratch-learned embedding technique. For the same experiment’s setting, the loss scores the lowest as 44.76%. When utilizing one of the stemming techniques—root stemming, ISRI, or Tashaphyne in scratch embedding—generally yields the best results. The highest accuracy of 82.72% is achieved with ISRI.

Experiments (B): static pre-trained word embeddings

It can take a while for the network to learn a word embedding, especially for very large text datasets. In this section, we will use pre-trained word embedding developed on a large text corpus. The first step is to load the word embedding as a word dictionary into vectors. The word vectors are positioned correctly in the weight matrix, which is built using the loaded embedding and the tokenizer word-index vocabulary as inputs.

In order to prevent the network from attempting to modify the pre-learned vectors as part of network training, observe that we set the ‘trainable’ argument to ‘False’ and provide the previously created weight matrix embedding-vectors as an argument to the fresh embedding layer. Figure 11 shows the results of ASA using deep learning methods with static pre-trained weights.

Figure 11 Results of 3-class sentiment classification using deep learning (CNN-static, BiLSTM-static, LSTM-CNN-static).

We used four word-embedding techniques (CBOW, Skip-gram, GloVe, FastText) with four stemmers (ISRI, root-based Tashaphyne, stem-based Tashaphyne, Snowball). Trainable = false, static weights.

According to the results of the static weights experiments, using a skip-gram with a snowball stemmer gives us the best accuracy at 83.6% when the LSTM-CNN classifier model is used. However, when the used classifier model is the standalone CNN, the original text without stemming worked well and was similar to Snowball light stemming results after we combined it with GloVe or fastText word embedding, where the accuracy scores were 82.4%. Using the BiLSTM classifier, the best accuracy was using Snowball or original text at 83.57%, and the lowest loss value was 44.93% with the GloVe.

Experiments (C): non-static pre-trained word embeddings

In this group of experiments, the weights in the embedding layer can serve as a network’s initial start and be modified as the network is trained. ‘trainable = True’ (the default parameter) can be used to achieve this when creating the embedding layer. Figure 12 shows the results of ASA using deep learning methods with the non-static pre-trained weights technique.

Figure 12 Results of 3-class sentiment classification using deep learning (CNN-non-static, BiLSTM-non-static, LSTM-CNN-non-static).

We used four word-embedding techniques (CBOW, Skip-gram, GloVe, FastText) with four stemmers (ISRI, root-based Tashaphyne, stem-based Tashaphyne, Snowball). Trainable = True, non-static weights.

Using Tashaphyne stems with CBOW embedding yields the best accuracy results for BiLSTM and LSTM-CNN models, at 83.44% and 84%, respectively, based on the findings of non-static weights. The BiLSTM model scores the lowest loss values with the fastText word embedding as 44.93%. Using the snowball stemmer and skip-gram with BiLSTM produced the best recall rates across all group experiments, reaching 84.38%.

Experiments (D): custom word embedding

Through this group experiments, we demonstrate how effectively we can acquire a standalone embedding that we can then apply to our neural network. The Word2Vec algorithm is a method for learning word embedding independently from a text collection. The method’s advantage is that it is capable of efficiently constructing high-quality word embedding concerning space as well as time complexity. First, the list of cleaned and preprocessed reviews from the training data is passed in. Then, we designated the dimensions of the embedding vector space to be (300). We take into account five neighbors when determining how to embed each word in the training reviews. Four threads were used during model fitting. We set the lowest count of occurrences for words to take into account in the vocabulary to one as we are learning new words. Figure 13 shows the results of ASA using deep learning methods with custom embedding weights technique.

Figure 13 Results of 3-class sentiment classification using deep learning (CNN-custom, BiLSTM-custom, LSTM-CNN-custom).

We used two word-embedding techniques (CBOW, Skip-gram) with four stemmers (ISRI, root-based Tashaphyne, stem-based Tashaphyne, Snowball).

The custom embedding technique indicates that we get superior accuracy results than the CBOW embedding when we use skip-gram with Tashaphyne-based roots. Specifically, we achieve 83.08%, 83.28%, and 83.05% using CNN, BiLSTM, and LSTM-CNN, respectively. The BiLSTM model here scores the lowest and the best loss value as 44.38%. We can notice that the BiLSTM model with Tashaphyne roots always performs accurately according to all group experiments in terms of loss value. Trainable outperforms scratch by 1.34 3.4%. When trainable = true is used, fastText accuracy decreases without utilizing a stemmer enhancement. CNN skip-gram used the Tashaphyne stemmer to obtain the best improvement, which was 3.5%. With ISRI stemmer improvement, trainable = true exhibits an accuracy improvement of up to 3%.

The accuracy results of the best experiment conducted by each group, scratch, static weights, non-static weights, and custom weights are shown side by side in Fig. 14. According to the results, CNN-LSTM had the lowest accuracy (79%) when utilizing bespoke weights without a stemmer, and the highest accuracy (84%) while using the Tashaphyne stemmer with non-static weights.

Figure 14 A summary of the accuracy results of 3-class sentiment classification using deep learning models and based on four different weighting techniques (scratch, static, non-static, and custom weights).

External evaluation and comparison

We used benchmark datasets, which were also used by other academics to evaluate their models, to compare our model with those of other researchers. Two sizable collections of reviews of books and hotels written in dialectical Arabic and Modern Standard Arabic (MSA) are called Hotels Arabic-Reviews Dataset (HARD) (Elnagar, Khalifa & Einea, 2018) and Books Reviews in Arabic Dataset (BRAD) (Elnagar & Einea, 2016). There are reviews and ratings out of five for each of them. By classifying ratings 1 and 2 as negative (−1), ratings 3 as neutral (0), and ratings 4 and 5 as positive (+1), we may convert the 5-star rating to 3-polarity scaling. Following rating conversion, there are 409,562 reviews total on HARD, divided into 276,387 positive, 52,849 negative, and 80,326 neutral reviews. There are 325,434 positive reviews, 78,380 negative reviews, and 106,785 neutral reviews out of the 510,599 total reviews on BRAD.

The confusion matrix presented in Fig. 15 shows the performance of the best model (LSTM-CNN) for multiclass classification. The normalized matrix evaluates the true positive rates towards each separated class. The area under the curve (AUC) is a single number that sums up a classifier’s performance by evaluating the ranking regarding the separation of each class as well. ROC is a probability curve that illustrates the degree or measure of separability by mapping the true positive rate (TPR), which represents the sensitivity vs. false positive rate (FPR), which represents specificity at various threshold levels. The receiver operating characteristic (ROC) is an excellent tool to visualize a classifier performance (Brownlee, 2023). Better still, the higher. The area under the ROC curve (AUC), which ranges from 0 to 1, represents the percentage of this region. The misclassification of neutral sentiments as positive or negative is a common issue in sentiment analysis models, especially in Arabic reviews. This is due to complex sentence structures and ambiguous language, which models struggle to interpret effectively. From both Figs. 15 and 16 we can see that the model operates worst with the neutral class, where only 13% of the actual neutral reviews were truly classified, while the remaining reviews were misclassified as either positive or negative. For example, the review was accidentally misclassified as positive; this may happen since it contains a positive word as .

Figure 15 Normalized confusion matrix of LSTM-CNN for multiclass classification.

Figure 16 ROC curve of LSTM-CNN for multiclass classification.

Figure 17 shows the normalized confusion matrix of LSTM-CNN for binary classification, Fig. 18 explains that the ROC curve becomes better when neutral reviews are excluded from the dataset, where the average AUC increases from 0.77 to 0.9.

Figure 17 Normalized confusion matrix of LSTM-CNN for binary classification.

Figure 18 ROC curve of LSTM-CNN for binary classification.

To establish a binary sentiment classification, neutral reviews were removed from the subsequent studies. We determined which experimentation settings worked best on the FDS and we found that skip-gram word embedding and stemming based on the Tashaphyne root produced the best results most of the time. Table 8 displays the results of the performance of each experiment.

Table 8 Performance evaluation on the external datasets.

Experiment
group	DL model	3-class classification	Binary classification	
HARD	BRAD	FDS	HARD	BRAD	FDS	
	Acc.	Loss	Acc.	Loss	Acc.	Loss	Acc.	Loss	Acc.	Loss	Acc.	Loss	
(A) scratch	CNN	88.2	31.5	80.3	35.8	83.2	46.6	96.4	10.4	81.5	29.9	90.6	26.5	
embeddings	BiLSTM	88.3	31.4	80.3	35.7	83.6	44.7	96.5	10.1	82.2	28.9	90.4	24.6	
	LSTM-CNN	88.1	32.1	80.2	35.9	83.5	45.2	96.2	11.2	82.4	28.2	89.2	30.3	
(B) static	CNN	88.5	30.7	80.4	35.6	82.4	47.4	96.6	10.2	82.8	27.5	90.5	24.8	
embeddings	BiLSTM	88.6	30.2	81.2	34.2	83.5	45.7	97.0	8.63	83.8	26.0	91.0	24.1	
	LSTM-CNN	88.4	30.9	81.3	34.3	83.6	45.9	96.3	10.3	83.9	25.5	91.0	24.0	
(C) non-static	CNN	88.5	30.6	81.0	34.4	83.5	45.2	97.0	9.04	81.9	29.5	91.7	21.9	
embeddings	BiLSTM	88.9	29.2	81.5	34.0	83.4	45.3	97.0	8.67	83.9	25.3	92.5	20.2	
	LSTM-CNN	88.9	29.5	81.5	34.0	84.0	45.4	96.2	10.3	83.8	25.6	92.5	21.1	
(D) custom	CNN	88.5	30.6	81.0	34.5	83.0	45.8	96.7	9.38	81.9	29.3	90.9	25.0	
embeddings	BiLSTM	88.9	29.3	81.2	34.2	83.3	44.3	97.2	8.31	82.0	29.5	91.3	22.6	
	LSTM-CNN	88.6	30.8	81.2	34.3	83.0	48.2	96.5	10.0	82.3	29.0	90.6	24.1	

The results we achieved were compared to those of a few other state-of-the-art deep learning models found in the literature. For Arabic SA, Nassif, Darya & Elnagar (2021) suggested Deep and Shallow Learning approaches. All conceivable hybrids of the three primary DL archetypes—CNN, LSTM, and GRU—, as well as all other archetypes, perform binary and 5-class classification. Their results when running binary classification reached: 94.2% and 83.5% respectively on HARD and BRAD.

The same corpora were utilized by Berrimi et al. (2023) to evaluate their hybrid bidirectional GRU/LSTM model for binary classification. They scored promising accuracies of 96.29% and 95.65% respectively on both datasets. Muaad et al. (2021) proposed ArCAR, a novel computer-aided deep-learning Arabic text recognition system that can analyze and interpret Arabic text at the character level. Each Arabic character in the Arabic text input is quantized into a 1D vector to create a 2D array for the ArCAR system. Five-fold cross-validation tests have approved the ArCAR system for two usages, Arabic text document categorization and Arabic sentiment analysis. The proposed deep learning ArCAR consists of two fully connected or dense layers and six convolutional layers for Arabic sentiment analysis. The model achieved the highest performance using the Arabic hotel reviews dataset (HARD) of two classes in terms of overall accuracy of 93.58%. The model employed the HARD and BRAD datasets for binary and 3-class sentiment classification. Additional studies that used the HARD and BRAD datasets provided by Habbat et al. (2023) for 3-class classification. They used a stacking deep learning model for classification, AraBERT for text representation, and AraGPT for data augmentation. Three deep learning models—GRU, LSTM, and Bidirectional LSTM (BiLSTM)—serve as the foundation classifiers for the suggested ensemble model. The accuracy for HARD and BRAD was 88.89% and 90.88%, respectively.

Table 9 summarizes the results of several state-of-the-art deep learning models compared to our proposed model. Comparing our results with previously described deep learning models, we scored the best binary classification results on HARD at 97.2%; our results on BRAD are comparable and close to some other models, such as the study of Nassif, Darya & Elnagar (2021). By comparing three classification outcomes, our results are also comparable and close to the model of Habbat et al. (2023) on the HARD dataset and are higher than the accuracy results of Muaad et al. (2021) on BRAD by approximately 4%.

Table 9 Comparison with some other state-of-the-art Deep Learning models for ASA.

Deep learning ASA model	HARD accuracy	BRAD accuracy	
Nassif, Darya & Elnagar (2021)	94.2% (binary)	83.5% (binary)	
Berrimi et al. (2023)	96.29% (binary)	95.65% (binary)	
Muaad et al. (2021)	93.58% (binary), 93.23% (3-class)	81.46% (binary), 77.13% (3-class)	
Habbat et al. (2023)	88.89% (3-class)	90.88% (3-class)	
Our model	97.2% (binary), 88.9% (3-class)	83.9% (binary), 81.5% (3-class)	

Conclusions and future work

This research developed a deep learning sentiment analysis model of reviews of food delivery services and online food ordering written in Arabic dialect during and after the global COVID-19 outbreak. We have effectively utilized extensive manual and automatic cleaning and preprocessing, as well as many Arabic text analysis techniques, including word embedding and variant stemming. Next, we used the FDS dataset to compare the performance of several well-known deep learning methods, including four-word embedding techniques, four stemmers, and three classifiers. We did this for binary and three-class classifications. We provided benchmarking data about the deep learning ASA literature and a comparison that highlights the similarities between our method results and those of other methods. Based on the superior findings of each standalone deep learning model, we plan to use multichannel and ensemble deep learning classifiers in the future.

Supplemental Information

Supplemental Information 1 Python source code for all experiments implementation.

Deep Learning Sentiment Analysis Static experiments, Scratch experiments, Non-Static experiments, custom experiments, the original dataset, and dialect dictionary.

Additional Information and Declarations

Competing Interests

The authors declare that they have no competing interests.

Author Contributions

Dheya Mustafa conceived and designed the experiments, performed the experiments, analyzed the data, performed the computation work, prepared figures and/or tables, authored or reviewed drafts of the article, and approved the final draft.

Safaa M. Khabour conceived and designed the experiments, performed the experiments, analyzed the data, performed the computation work, prepared figures and/or tables, authored or reviewed drafts of the article, and approved the final draft.

Mousa Al-kfairy analyzed the data, authored or reviewed drafts of the article, and approved the final draft.

Ahmed Shatnawi analyzed the data, authored or reviewed drafts of the article, and approved the final draft.

Data Availability

The following information was supplied regarding data availability:

The talabat reviews dataset was obtained from talabat.com.

The data, annotated to 3 class classification (pos,neg,nue) by native Arabic speakers, is available at Mendeley: Mustafa, Dheya (2023), “Talabat”, Mendeley Data, V1, DOI: 10.17632/gpg5s6g8s7.1.

The code is available in the Supplemental Files.

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
