# Peer review of "Leveraging sentiment analysis of food delivery services reviews using deep learning and word embedding"

_PeerJ Computer Science, doi:10.7717/peerj-cs.2669_

## Round 0.1 · original submission · Major Revisions

· Academic Editor

Major Revisions

Based on both reviewers' comments, the manuscript requires major revision primarily due to significant methodological gaps, poor documentation, and validation issues. The key problems include: unclear embedding techniques and model fine-tuning processes, missing dataset characteristics, incomplete literature review (especially recent works), lack of state-of-the-art comparisons, and inconsistent terminology.

All methodology-related comments must be addressed, along with requests for flowcharts and dataset documentation, while suggested references can be added at the authors' discretion.

The reviewers missed some critical points like model validation procedures, error analysis, code availability, and potential biases in the Arabic dataset.

Despite these issues, the manuscript shows promise in its focus on Arabic sentiment analysis and embedding comparison approach, making it suitable for major revision rather than rejection. For improvement, the authors should prioritize documenting their methodology clearly, expanding their literature review, adding proper validation comparisons, and maintaining consistent terminology throughout the paper.

·

Basic reporting

The manuscript presents several significant issues that need to be addressed before it can be considered for publication. Please find my detailed comments and concerns below:
1. Please add two paragraphs in the introduction: a) objectives and motivations tied to gaps in the literature; b) research questions.
2. The literature review of the article doesn't provide comprehensive coverage of related work. Most troublesome is the significant gap in coverage of the state-of-the-art. A few important pieces of literature about deep learning are not cited.

3. It is not clear how the authors used the embedding techniques. Please provide more explanations.
4. It is not clear how the authors fine-tuned the model. Please add experiments and details.
5. It is not clear how the parameters of the models are compared.
6. I would strongly advise including all major works in 2023 and 2024 and drawing a tabular comparison among your work and other works. A few works worth comparing, referring to, or citing are below:

(a). Sentiment analysis on images using different transfer learning models, DOI: https://doi.org/10.1016/j.procs.2023.01.142
(b). Point of Interest Recommendation System Using Sentiment Analysis, DOI: https://doi.org/10.1633/JISTaP.2024.12.2.5

7. There is no such sub-section mentioning the details about the dataset size. I couldn’t find it anywhere regarding the size or quantity of the dataset.
8. I'd recommend adding some possible improvements for the proposed approach.
10. What are the avenues for future research or improvements identified based on the findings of this study?
11. The challenges in the work need to be stated (As mentioned in the Title)
12. What are the challenges in the dataset?

Experimental design

7. There is no such sub-section mentioning the details about the dataset size. I couldn’t find it anywhere regarding the size or quantity of the dataset.
8. I'd recommend adding some possible improvements for the proposed approach.

Validity of the findings

10. What are the avenues for future research or improvements identified based on the findings of this study?
11. A comparison with SOTA needs to be included.

Reviewer 2 ·

Basic reporting

This manuscript, "Leveraging Sentiment Analysis of Food 2 Delivery Services Reviews Using Deep 3 Learning and Word Embedding," tried to identify aspects where the customer experience of food delivery companies (meal delivery services or FDS) can be improved by Using a Modern Standard Arabic and Dialect Arabic corpus dataset collected from Talabat.com, an RNN-based sentiment analysis classification model. The proposed model is hard to appreciate because many parts of the paper are unclear. However, despite all this complexity, the comparison of different word embedding vectors in the experimental results and the obtained performance are promising. However, despite all this complexity, comparing different word embedding vectors in the experimental results and the obtained performance gives hope that the article can be improved with a good revision process.

Experimental design

*More detailed explanation of the proposed method is required with the computational complex analysis.
- Authors may compare their work with existing literature.
*There is no clear description of the model used to calculate sentiments. How was it trained? What is the reference for the model?
*In the study, word clouds can be used for text visualization. Such a visualization is necessary to understand the dataset and its content.
-Wordcloud representations before and after preprocessing
-Specific word clouds of the detected sentiment classifications
*A flowchart showing all the steps performed, including preprocessing, should be included in the manuscript to support a detailed representation of the proposed model.

Validity of the findings

*When the findings obtained with the model proposed in the article are evaluated, the results are promising. Critical explanations can be expanded in the discussion section to emphasize the superiority of the proposed model.
*They stated that they applied emotion analysis techniques in some places in the text and sentiment analysis techniques in other places. One must be consistent when using such definitions.

---

## Round 0.2 · Minor Revisions

· Academic Editor

Minor Revisions

The reviewers have indicated that the authors have adequately addressed their comments, and the manuscript has seen substantial improvements compared to its initial submission. However, they have also highlighted a few minor concerns that should be resolved before the manuscript can be considered for acceptance.

·

Basic reporting

The author has incorporated all the changes.

Experimental design

The author has incorporated all the changes.

Validity of the findings

The author has incorporated all the changes.

Additional comments

The author has incorporated all the changes.

Reviewer 2 ·

Basic reporting

The authors have responded to all my comments. The paper had significant improvement compared to the first version. However, there are some small concerns that need to be addressed before accepting the manuscript.

Experimental design

• The original dataset used in the article does not have sentiment labels. Which technique is used to detect sentiment labels in the study?
• How was the sentiment polarity calculated in the technique used? Explanatory information should be provided.

Validity of the findings

It has been improved with revision.

---

## Round 0.3 · accepted · Accept

· Academic Editor

Accept

The improvements made in the final revision indicate that the article is now in an acceptable for publication.